

# Internet risky behaviours among youth with visual impairment

Magdalena Agnieszka Wrzesińska[1], Kamila Knol-Michałowska[2], Patryk Stecz[3], Monika Kopytowska[4] and Katarzyna Binder-Olibrowska[5]

[1] Department of Psychosocial Rehabilitation, Medical University of Lodz, Lodz, Poland
[2] National Centre for Workplace Health Promotion, Nofer Institute of Occupational Medicine, Lodz, Poland
[3] Department of Clinical Psychology and Psychopathology, Institute of Psychology, Faculty of Educational Sciences, University of Lodz, Lodz, Poland
[4] Department of Pragmatics, Institute of English Studies, Faculty of Philology, University of Lodz, Lodz, Poland
[5] Department of Psychosocial Rehabilitation, Medical University of Lodz, Lodz, Poland

Corresponding author
Magdalena Agnieszka Wrzesińska,
magdalena.wrzesinska@umed.lodz.pl

## ABSTRACT

**Background:** Young individuals with visual impairment (VI), exposed to higher risky of social exclusion, might be more prone to Internet risky behaviours including electronic aggression.

**Objective:** Different types of Internet risky behaviours and the average time spent online were investigated among students with VI. These behaviours were analyzed for the relationship with witnessing, perpetrating and becoming a victim of electronic aggression.

**Methods:** A total of 490 special needs school students with VI answered a self-administered questionnaire. The average time spent online, different types of risky Internet activities and electronic aggression were recorded, as well as sociodemographic characteristics.

**Results:** Male students downloaded software illegally, hacked, published and viewed sites with sexual content, and gambled online statistically more often than female counterparts. Concerning electronic aggression, more than half of participants were engaged as witnesses, every fifth student as a victim and 11.6% as perpetrators. Two sets of ordinal logistic regression analyses have shown that communication with a person promoting violence and an excessive amount of time spent online during weekends predicted significantly the risk of witnessing and being a victim of electronic aggression. Additionally, communicating with a person promoting violence and an excessive amount of time spent online during schooldays were associated with increased odds for engagement in electronic aggression as a perpetrator.

**Conclusions:** Population with VI is prone to risky Internet use and likely to be engaged in electronic aggression. New instruments and preventive strategies need to be developed, as well as diagnostic tools tailored specifically to the needs of young people with VI.

## INTRODUCTION

Online activities play a crucial role in young people's daily life, transforming, both qualitatively and quantitively, the ways in which they socialize and learn (*Bosse, Renner & Wilkens, 2020*; *Livingstone, Mascheroni & Staksrud, 2018*). One of the most important reports on youth's online activities has been developed as part of the EU Kids Online project. The main aim of this project is to provide high quality, independent and comprehensive research findings to improve the use and security of an online environment for young people. Surveys have been regularly conducted among school students for over a decade by research teams from 34 countries applying an integrated research expertise across multiple disciplines and methods. The project provides an opportunity to map youth's experience of online spaces including social and cultural aspects and it delivers the necessary information regarding cyberspace for practitioners as well as the national and international policy stakeholders (*EU Kids Online IV Survey, 2019*).

According to the results of the last Polish EU Kids Online research almost half of young people use the Internet for up to 2 h on schooldays and for up to 3 h during weekends. Using social networking services (SNS) and video/audio conferencing tools, playing games, watching movies and listening to music are done most often every or almost every day by students (*Pyżalski et al., 2019*). The Internet, with multiple websites and applications allowing for a quick exchange of information, has become a learning tool and platform. As demonstrated by the EU KIDS report, 60% of school students use the Internet for doing homework, preparing for tests, and expanding knowledge needed for school (*Pyżalski et al., 2019*).

There are, however, also negative aspects associated with Information and Communication Technologies (ICT) and digital communication. *Tahiroglu et al. (2008)* argue that the Internet has a negative impact on young people's psychosocial development due to the fact that it is time-consuming and uncontrollable. As posited by social compensation theory, negative life events can enhance motivation to go online to relieve negative feelings (*Valkenburg & Peter, 2009*). In this context, the COVID-19 pandemic and the resulting spatial distancing, along with the introduction of distance learning solutions, have caused anxiety, thus leading to excessive Internet use (*Király et al., 2020*). Despite enabling various forms of mediated interactions, information exchange, as well as constructive and collaborative activities, cyberspace with its technological affordances has also become an environment where harmful and violent content of all kinds can be accessed, generated, amplified and distributed (*KhosraviNik & Esposito, 2018*; *Kopytowska, 2017*).

Electronic aggression is indeed one of the most important online phenomena to which young Internet users are exposed. It is most often defined as actions which have been carried out against the will of victims and caused social, psychological or moral harm. The concept of electronic aggression is broad and it is significantly different from cyberbullying. The only thing that is common to these two concepts is the use of ICT. The cyberbullying criteria, such as repeatability, relation between a victim and a perpetrator (in cyberbullying it is the same social group, for example school class, whereas

in case of electronic aggression it is not specified—it can be both: people that a perpetrator knows and strangers), aim of harming the victim and imbalance of power do not apply in the case of electronic aggression. In addition, electronic aggression can be used both in a situation where an aggressive action is made directly against another person (*e.g.* sending an offending e-mail), and when an action is indirect (*e.g.* compromising materials are published on the Internet) (*Pyżalski, 2009*). A recent Polish study showed that among those who faced aggression off-line about 30% of youth have never been affected by aggression *via* the Internet, and also about every third person aged 9–17 has never been a perpetrator of electronic aggression. On the other hand, about 14% of older students have received sexual messages and 7% of them have been perpetrators of cyberbullying or hate speech (*Pyżalski et al., 2019*).

As posited by Media Proximization Approach (MPA) the transgression of spatial and temporal boundaries made possible thanks to the techno-discursive design of digital media (*i.e.* technological aspects along with presumption-related discursive practices) (*KhosraviNik, 2018*) not only enables and facilitates user-to-user interaction, but also allows users to access, generate and share media content (*Kopytowska, 2013*; *Kopytowska, 2015*; *Pyżalski, 2014*; *Kopytowska, 2020*). Interactivity and connectivity translate into greater emotional involvement. Technically, the Internet and social media sometimes encourage the spread of emotionally loaded content in a way that cannot be controlled (*KhosraviNik & Esposito, 2018*). Anonymity is a key factor here. The Social Identity Model of Deindividuation Effects (*Postmes, Spears & Lea, 2002*) points to serious cognitive consequences of anonymity in online interactions including various forms of anti-social behaviour, such as flaming (posting insults and other offensive language), trolling (posting inflammatory, digressive and off-topic messages in a particular online community— forum, chat room, blog—with the intention of provoking its members to respond in an emotional way) and other forms of online incivility (various forms of offensive interaction including aggressive comments, hate speech and harassment) (*Santana, 2014*). Deindividuation experienced due to perceived anonymity may thus enhance disinhibition: individuals act like they would never do in face-to-face communication (*Pyżalski, 2014*). MPA argues that technologically-enabled changes in distance dynamics (involving interactivity patterns and anonymity), make potential victims of online aggression more accessible to perpetrators and vulnerable to the attacks (*Kopytowska, 2020*). Likewise, they also make young Internet users more prone to online risky behaviours, in terms of both exposure and performance. Mubarak classifies risk-taking behaviours into three categories, namely dangerous interactions with people, accessing dangerous websites and other risk-taking behaviours (*Murabak, 2012*). One of the most comprehensive typologies of online risky behaviours was developed for the EU KIDS Online research (*Livingstone, Mascheroni & Staksrud, 2018*; *Hasebrink et al., 2011*) and adapted by Polish authors (*Pyżalski, 2012*). It includes two dimensions—types of online activities (content-based activities, contact- or communication-based activities, and conduct or peer-participation activities) and types of risk (aggressive, sexual, values, and commercial). Numerous other studies investigated youth in the context of cyber aggression or cyberbullying (*Kowalski et al., 2014*; *Tsitsika et al., 2015*), sexually explicit content (*Beyens, Vandenbosch &*

*Eggermont, 2015*; *Martellozzo et al., 2020*), and self-harm activities (*Villani, Florio & Castelli, 2015*). The factors that affect susceptibility to online risky behaviours included age, gender, parental control, as well as parents' education level. Study results showed that girls, older teenagers, and those whose parents have low education experience risky online behaviours more frequently (*Tsitsika et al., 2012*; *Anderson, 2018*). Additionally, girls are more frequently exposed to information about self-harming, ways of committing suicide or extreme weight loss, while boys more often search for online pornography (*Martellozzo et al., 2020*; *Tsitsika et al., 2012*). The Internet use among people with disabilities is correlated with age and more likely to occur among young people. Youth with a physical disability are excluded from participation in the society because of restrictions in daily life, such as self-care, reduced mobility, or communication activities. Hence, a number of studies discuss the need for Internet use among young people with disabilities (*Argen, Kiellberg & Hemminsson, 2020*; *Parimala et al., 2012*). The Internet has considerably empowered this group, thus enhancing the degree of its participation in the social life (*Duplaga, 2017*). It has become the main e-resource for students with VI because online activities give them an opportunity to learn and work independently, when compared to traditional methods (*Pfeiffer & Pinquart, 2013*). It should be noted, however, that efficient and independent use of the Internet by people with VI often requires additional optical equipment, hardware, and software in the form of screen readers, magnifiers, or Braille monitors. Websites and mobile applications should comply with Web Content Accessibility Guidelines (WACG), thus making the use of the Internet more user-friendly and functional. Such solutions make it easier for people with VI to adjust page settings (*e.g.* high contrast, larger font) to their needs in order to see the content better (both words and images). Still, the adaptation of technological devices requires understanding of the specific problems of end-users whose feedback is essential to develop personalized aids for the blind and poor-sighted (*W3C Web Accessibility Initiative, 2005*).

Young people with chronic health diseases and disabilities are more likely to become victims of peer harassment (*Melissa et al., 2019*). Previous Polish studies focused on the statistical relationship between cyber-victimization and having problems with visible bodily injury and disability. Male students with health problems were significantly more exposed to being a victim of both traditional bullying and cyberbullying (*Plichta, Pyżalski & Barlińska, 2018*). Negative body image, reduced self-esteem, decreased social and communication skills or poor academic performance have been identified as the main predictors of being the target of victimization in this group (*Pinquart, 2017*). Although, the prevalence rate of bullying depends on illness or disability, the highest risk of victimization was observed among youth with VI (*Pinquart, 2017*). Another preliminary study (*Wrzesińska, Tabała & Stecz, 2016*) also confirmed that students with VI spent more time on online activities compared to their peers without disabilities. About three quarters of them used the Internet for learning purposes and over 60% spent their free time in social media. They were also prone to on-line risky behaviours, with boys being more likely to download files illegally, hack or watch pornography than girls.

However, the above-mentioned research was conducted among a small number of students with disability and provides no details as regards the correlation between the

specific online risky behaviours and different types of electronic aggression. To fill the information gap in this area it seems necessary to understand the motivations behind both risky behaviours and online aggression among young people who are blind or poor-sighed (*Plichta, Pyżalski & Barlińska, 2018*; *Wrzesińska, Tabała & Stecz, 2016*). Research in this area is essential and highly relevant, especially when an offensive type of communication is correlated with stereotypes including various social characteristics such as gender, religion, age, race or disability (*Plichta, Pyżalski & Barlińska, 2018*; *Chetty & Alathur, 2018*). For this reason, it is important to find answers to the main research questions: What is the frequency and nature of Internet use among young people with VI? Are students with VI involved in online risky behaviours? What kind of relations between online behaviours and manifestations of electronic aggression (victim/witness/perpetrator) can be identified among young people with VI?

The following hypotheses have been formulated:

a) Male students with VI spend more time online and vary in the frequency of Internet activities compared to female counterparts.

b) Involvement in specific Internet activities is characterized by a more frequent use of the Internet for (i) learning purposes and (ii) social media than the use of forums, blogs, and online shopping among young people with VI.

c) Male students with VI engage in particular online risky behaviours (downloading software illegally without a valid license, hacking and accessing pornographic content) more frequently than female students.

d) Students with VI are more likely to be victims than perpetrators of electronic aggression.

e) Online risky behaviours and time spent on the Internet are positive predictors of all forms of engagement in electronic aggression (victim, perpetrator, witness) among young people with VI.

## MATERIALS AND METHODS

### Selection procedure

The sampling procedure comprised several stages. First, schools were randomly selected from among nine Polish special-needs schools for the blind and partially sighted. All students aged 13–24 years old from the lower-secondary and upper-secondary classes were invited to participate in the study. The age threshold of our group is related to the fact that Polish students with disabilities have the possibility to extend their education time up to 24 years of age. The inclusion criteria for students were as follows: confirmation of VI according to the International Classification of Diseases (ICD-10, 2014), presence during the day of data collection and formal consent to take part in the study, also from parents in the case of students under 18. Parents and participants received written information about the study with the consent form before the survey. The approved and signed consent in the paper form had to be delivered to persons supervising the administrative course and it was a prerequisite for inclusion in the study. To make sure

that all participants with VI understood the conditions of the survey, the information about the project and the possibility of withdrawing from the survey at any time was presented before the survey. This information was written in Braille, large font and read aloud to all respondents just before the survey. The only exclusion criterion was the presence of a diagnosed physical or mental disability other than VI. The project was approved by the Bioethics Committee of the Medical University of Lodz (No. RNN/802/14/KB).

## Participants

A total of 1,018 respondents met the criteria of participation in the survey. The withdrawal from research during implementation, lack of data, as well as failure to obtain consent for the participation were the main reasons why 490 respondents with VI (boys: 259; girls: 231), aged 13–24 years (17.9 ± 2.48) were recruited. The students who admitted to using the Internet were blind ($N = 70$; 14.3%) or partially sighted ($N = 420$; 85.7%). Additional information regarding the severity of impairment or socioeconomic background was not collected as it is considered sensitive data in Poland. The participants comprised two subgroups: lower-secondary students ($N = 157$; 32.0%) and upper-secondary students ($N = 333$; 68.0%).

## Measurement tool

The survey was conducted under the guidance of a trained team: a public health specialist and a psychologist; caregivers and teachers were absent during the survey. A self-administered questionnaire was created and prepared in Braille or large font for the needs of this study. In the questionnaire, questions on sociodemographic variables included gender, age (birth year) and type of school. The first section recorded mean time of Internet sessions including time spent during schooldays and weekends in minutes. Time spent on online activities on computers and mobile devices was estimated on the basis of participants' declarations. In the next section, frequency of online activities, such as communicating *via* the Internet, SNS, discussion forums, downloading of MP3, music or software files, e-mails, searching for information for learning purposes, online shopping, blogs were considered when the nature of the Internet use was described (*Wrzesińska, Tabała & Stecz, 2016*). The response options to ten items indicating online activities on the Likert scale ranged from: never/almost never, sometimes, almost always/always. The third part of the questionnaire comprised seven items on a five-point scale about types of Internet activities involving risky behaviours (downloading software illegally without a valid license, publication of photos and movies on YouTube without permission of their owners, hacking, communication with a person promoting violence, posting information about sex, viewing websites with sexual content and gambling online) which were taken into consideration (*Wrzesińska, Tabała & Stecz, 2016*). The final section was devoted to electronic aggression involvement. Our primary objective was to identify and describe tendencies regarding electronic aggression among young people with three roles: perpetrators, victims, and witnesses. A perpetrator was defined as a person who had threatened or offended another person *via* the Internet. Victims were persons who had been threatened or offended by another person and a witness was a person who

had been exposed to the situation in which someone had been threatening or offending others *via* the Internet (*Jenaro et al., 2018*). This section used a five-point Likert scale (never, almost never, sometimes, almost always, always) about perpetrating, being the victim or witnessing electronic aggression.

## Statistical analyses

The nonparametric Mann–Whitney *U*-test was used to compare subgroups with regard to the amount of time spent online. The assessment of the frequency of risky behaviours and ways of using the Internet was made using the chi-square test of independence. In the case of fewer than five cases the Yates correction was included. To determine whether electronic aggression (being a perpetrator, a victim, or a witness) could be predicted by risky Internet behaviours, three separate ordinal logistic regression analyses were conducted. The criterion for inclusion of the predictors in the final logistic regression models was based on performing initially separate logistic regression equations with one explanatory variable. Based on the significance level ($p < 0.05$), the predictor was either used in the final model with multiple predictors or excluded from further analyses ($p < 0.05$). *P*-value lower than 0.05 was considered to be significant.

## RESULTS

### The frequency and nature of Internet use

The mean time dedicated to Internet use was longer during school-free days compared to schooldays (193.0 min ± 222.9 vs 88.3 min ± 123.8; $z = 14.414$; $p = 0.000$) in the total group. Over 60% ($N = 299$) of all participants were online up to 2 h on schooldays, and almost 66% ($N = 323$) of them usually spent up to 3 h at weekends. Male students spent more time on the Internet than female students both on schooldays (97.8 min ± 140.9 vs 77.7 min ± 100.7) and at weekends (207.2 ± 234.4 min vs 176.6 min ± 208.1). Students from upper-secondary schools spent more time on the Internet than those from lower-secondary schools on schooldays (96.6 ± 135.2 min vs 70.0 min ± 91.7) and during school-free days (202.4 min ± 209.3 vs 172.2 min ± 228.4). Half of the students used the Internet always or almost always for social networking or to obtain information needed for school. Female youth used SNS statistically more often than male youth, but male students downloaded MP3, software or music files or commented on discussion forums statistically more often than females. There were no statistical differences between lower and upper-secondary school students with regard to almost all types of Internet use. The only difference concerned the fact that students from upper-secondary schools read or sent more e-mails ($N = 66$; 20.8% vs $N = 15$; 9.9%; Chi$^2$ = 14.325; $p = 0.0008$, respectively) and they also statistically more often used computers to obtain information for knowledge development ($N = 176$; 55.3% vs $N = 47$; 31.3%; Chi$^2$ = 23.38; $p = 0.00001$) than students from lower-secondary schools (Table 1).

### Online risky behaviours

In the sample, downloading music files and software without a license ($N = 161$; 34.2%) and viewing online sexual sites ($N = 69$; 14.4%) were most frequently indicated as

**Table 1 Types of Internet activity among young people with visual impairment depending on gender.** The row data and percentages are included in the table.

| | Total | | | Boys | | | Girls | | |
|---|---|---|---|---|---|---|---|---|---|
| | Never/ almost never N (%) | Sometimes N (%) | Always/ almost always N (%) | Never/ almost never N (%) | Sometimes N (%) | Always/ almost always N (%) | Never/ almost never N (%) | Sometimes N (%) | Always/ almost always N (%) |
| Conversation *via* the Internet (chats, Skype) | 125(26.9) | 204(43.9) | 136(29.2) | 59(23.8) | 117(47.2) | 72(29.0) | 66(30.4) | 87(40.1) | 64(29.5) |
| SNS | 85(18.0) | 136(28.9) | 250(53.1) | 55(22.1) | 79(31.7) | 115(46.2) | 30(13.5) | 57(25.7) | 135 (60.8)[1] |
| Discussion forums | 288(61.5) | 145(31.0) | 35(7.5) | 138(55.6) | 84(33.9) | 26(10.5) | 150(68.2) | 61(27.7) | 9(4.1)[2] |
| MP3, music and software downloading | 101(21.5) | 198(42.0) | 172(36.5) | 55(22.1) | 87(34.9) | 107(43.0) | 46(20.7) | 111(50.0) | 65(29.3)[3] |
| e-mails | 179(38.2) | 208(44.5) | 81(17.3) | 91(36.7) | 109(44.0) | 48(19.3) | 88(40.0) | 99(45.0) | 33(15.0) |
| Searching for information for learning purposes | 35(7.5) | 210(44.9) | 223(47.6) | 18(7.3) | 123(49.6) | 107(43.1) | 17(7.7) | 87(39.6) | 116(52.7) |
| Online shopping | 303(64.3) | 143(30.4) | 25(5.3) | 160(64.3) | 73(29.3) | 16(6.4) | 143(64.4) | 70(31.5) | 9(4.1) |
| Blogs | 344(73.2) | 88(18.7) | 38(8.1) | 179(72.2) | 50(20.2) | 19(7.6) | 165(74.3) | 38(17.1) | 19(8.6) |

Notes:
[1] $Chi^2 = 11.00$; $p = 0.004$.
[2] $Chi^2 = 10.77$; $p = 0.005$.
[3] $Chi^2 = 12.46$; $p = 0.002$.

performed sometimes or regularly. Male students engaged in all risky behaviours more often than female students. They statistically more often downloaded software illegally and hacked, published, and viewed sites with sexual content and gambled online (Table 2). Students from upper-secondary schools only downloaded software illegally always/almost always statistically more often than those from lower-secondary schools ($N = 32$ (9.9%) vs $N = 11$ (74%); $Chi^2 = 12.596$; $p = 0.0018$).

## Electronic aggression

In our sample almost 52% ($N = 241$) of students witnessed someone experiencing some form of violence on the Internet. Every fifth student ($N = 98$) experienced being a victim, and almost 12% ($N = 56$) of them were perpetrators of threatening or offending others on the Internet. However, no statistical differences were identified in this aspect. Although male youth had higher scores in every dimension of aggression more often than female youth, there were no statistically significant differences related to gender (Table 3). There were also no statistical differences between students from lower- and upper-secondary schools regarding the frequency of being a witness, a victim or a perpetrator of electronic aggression.

## Ordinal logistic regression models explaining electronic aggression

In order to identify the predictors of electronic aggression (witnessing), the following variables were included in multiple logistic regression: downloading software illegally without valid license, hacking, communication with a person promoting violence, viewing

**Table 2 Online risky behaviours and gender.** The raw data and percentages are included in the table.

| Risky behaviours | Total | | | Boys | | | Girls | | |
|---|---|---|---|---|---|---|---|---|---|
| | Never/ almost never N (%) | Sometimes N (%) | Always/ almost always N (%) | Never/ almost never N (%) | Sometimes N (%) | Always/ almost always N (%) | Never/ almost never N (%) | Sometimes N (%) | Always/ almost always N (%) |
| Downloading the software illegally without valid license | 310 (77.2) | 118(25.1) | 43(9.1) | 144(57.8) | 66(26.5) | 39(15.7) | 166(74.8) | 52(23.4) | 4(1.8)[1] |
| Publication of photos and movies on YouTube without permission of their owners | 433 (90.8) | 34(7.1) | 10(2.1) | 223(88.5) | 22(8.7) | 7(2.8) | 210(93.3) | 12(5.3) | 3(1.3) |
| Hacking | 417 (87.4) | 42(8.8) | 18(3.8) | 201(79.8) | 34(13.5) | 17(6.7) | 216(96.0) | 8(3.6) | 1(0.4)[2] |
| Communication with persons promoting violence | 433 (90.6) | 31(6.5) | 14(2.9) | 225(88.9) | 17(6.7) | 11(4.3) | 208(92.5) | 14(6.2) | 3(1.3) |
| Publication of information regarding sex | 445 (93.3) | 18(3.8) | 14(2.9) | 227(90.1) | 13(5.2) | 12(4.7) | 218(96.9) | 5(2.2) | 2(0.9)[3] |
| Viewing sexual pages online | 409 (85.6) | 56(11.7) | 13(2.7) | 198(78.6) | 41(16.3) | 13(5.1) | 211(93.4) | 15(6.6) | 0(0.0)[4] |
| Gambling online | 440 (92.4) | 27(5.7) | 9(1.9) | 226(89.7) | 17(6.7) | 9(3.6) | 214(95.5) | 10(4.5) | 0(0.0)[5] |

Notes:
[1] $Chi^2 = 30.26$; $p = 0.000$.
[2] $Chi^2 = 29.42$; $p = 0.000$.
[3] $Chi^2 = 9.38$; $p = 0.009$.
[4] $Chi^2 = 24.14$; $p = 0.001$.
[5] $Chi^2 = 9.52$; $p = 0.008$.

**Table 3 The frequency of electronic aggression (witness, victim, perpetrator) in e-mails, or in social media.** The raw data and percentages are included in the table.

| Behaviours | Boys | | | Girls | | |
|---|---|---|---|---|---|---|
| | Never/ almost never N (%) | Sometimes N (%) | Always/ almost always N (%) | Never/ almost never N (%) | Sometimes N (%) | Always/ almost always N (%) |
| Have you ever witnessed someone threatening or offending other people *via* the Internet? | 119(49.4) | 106(44.0) | 16(6.6) | 105(46.9) | 111(49.5) | 8(3.6) |
| Have you ever been threatened or offended by another person *via* the Internet? | 196(77.2) | 49(19.3) | 9(3.5) | 188(82.5) | 37(16.2) | 3(1.3) |
| Have you ever threatened or offended another person *via* the Internet? | 220(86.6) | 30(11.8) | 4(1.6) | 205(90.3) | 21(9.3) | 1(0.4) |

sexual content online and amount of time spent on the Internet on schooldays and at weekends. It was shown that two out of four variables had significant predictive power for distinction between witnessing or non-witnessing electronic aggression. Those included: communication with a person promoting violence and the amount of time spent on the Internet during school or school-free days. It was shown that those who communicated occasionally with a person promoting violence were three times more likely to be engaged in electronic aggression as a witness when compared to those who did not have such experience (OR = 3.07; $p < 0.05$). No significant relationship was found between
**Table 4 Ordinal logistic regression models explaining electronic aggression (witness, victim, perpetrator) with multiple predictors.** The raw data and percentages are included in the table.

| Witness | OR | 95%CI | P |
|---|---|---|---|
| Communication with the person promoting violence | | | |
| Never/almost never | 1.00 | Ref. | |
| Sometimes | 3.07 | [1.29–7.32] | 0.011 |
| Always/almost always | 1.97 | [0.44–8.87] | 0.374 |
| Time of online activity during school-free days (in h) | | | |
| 0 | 1.00 | Ref. | |
| Less 1 | 1.91 | [0.78–4.66] | 0.155 |
| From 1 to 2 | 1.69 | [0.70–4.08] | 0.241 |
| From 2 to 4 | 1.97 | [0.81–4.79] | 0.135 |
| Over 4 | 3.81 | [1.44–10.1] | 0.007 |
| **Victim** | | | |
| Communication with the person promoting violence | | | |
| Never/almost never | 1.00 | Ref. | |
| Sometimes | 3.26 | [1.40–7.60] | 0.006 |
| Always/almost always | 5.52 | [1.37–22.3] | 0.016 |
| Time of online activity during school-free days (in h) | | | |
| 0 | 1.00 | Ref. | |
| Less 1 | 2.60 | [1.23–5.51] | 0.012 |
| From 1 to 2 | 3.23 | [1.50–6.98] | 0.003 |
| From 2 to 4 | 2.57 | [1.07–6.13] | 0.034 |
| Over 4 | | | |
| **Perpetrator** | | | |
| Communication with the person promoting violence | | | |
| Never/almost never | 1.00 | Ref. | |
| Sometimes | 4.07 | [1.61–10.3] | 0.003 |
| Always/almost always | 3.32 | [0.67–16.5] | 0.142 |
| Time of online activity during school days (in h) | | | |
| 0 | 1.00 | Ref. | |
| Less 1 | 1.57 | [0.57–4.29] | 0.379 |
| From 2 to 3 | 3.05 | [1.07–8.710] | 0.036 |
| Over 3 | 4.02 | [1.23–13.1] | 0.021 |

communicating frequently with a person promoting violence and being exposed to electronic aggression as an observer (OR = 1.97, $p > 0.05$). It turned out—the ratio was approximately four to one—that the excessive Internet use during weekends (over 4 h per day) would be associated with witnessing electronic aggression (OR = 3.81; $p < 0.01$). Spending between 2 and 4 h online per day during weekends did not significantly affect witnessing electronic aggression (OR = 1.97, $p < 0.05$). The ratios for other variables were not significant (Table 4).

In the second ordinal regression model it was determined which independent variables had a statistically significant effect on being engaged in electronic aggression as a victim. The ratio for students communicating occasionally with those who promoted violence and, in this way, becoming the victims of electronic aggression was 3.26 ($p < 0.01$) of that for non-communicating students. The odds were higher among those who communicated very often (OR = 5.52, $p < 0.05$). Engagement in electronic aggression as a victim was almost three times more frequent if participants were online 1 h per day during school-free days. The odds were higher among the participants who used the Internet between 2 and 4 h per day during weekends (OR = 3.23, $p < 0.01$) and when the screen time was more than 4 h (OR = 2.57, $p < 0.05$, respectively). Other variables did not explain the engagement in electronic aggression as a victim (Table 4).

In the final ordinal regression model, it was determined that engagement in electronic aggression as a perpetrator was explained by communicating with a person promoting violence and time online during school days. Participants who responded "occasionally" with reference to communicating with a person promoting violence were engaged in electronic aggression as perpetrators over four times more often than non-communicating participants. The odds were lower and not significant among those reporting to communicate "frequently" with a person promoting violence (OR = 3.32, $p > 0.05$). An increase in Internet time (2–3 h per day during school days and more than 3 h vs less than 2 h) were associated with higher odds of being engaged in electronic aggression as a perpetrator (OR = 3.05, $p < 0.05$ and 4.02, $p < 0.05$, respectively). No significant odds increase or decrease were found for other independent variables explaining an engagement in electronic aggression as a perpetrator (Table 4).

## DISCUSSION

Our results showed that the mean time spent on the Internet during school-free days was over twice as long as during schooldays. Male students were online longer than female students both during schooldays and school-free days. These results confirmed our primary hypothesis. The tendency identified is consistent with the results of the previous study conducted recently among Polish adolescents from mainstream schools (*Pyżalski et al., 2019*). It should also be noted that the mean time dedicated to online activities increases with age among participants of this study. We could predict that it is connected with greater focus on learning activities or higher digital competencies among older students with VI.

Our study also confirmed that the Internet plays a very important role in knowledge development among students with VI. Indeed, access to e-resources is crucial for self-development and independence during school time and in adulthood among the members of this group. The online activities were mainly used for social networking or getting information to develop school knowledge, which confirms our second hypothesis. These results are very close to the previous preliminary study conducted among students with VI (*Wrzesińska, Tabała & Stecz, 2016*). Another study revealed that using online chat had a positive impact on the well-being of people with VI but searching for disability-related information or participation in online support groups influenced them

negatively (*Smedema & McKenzie, 2010*), which may be explained by the stress built up during information search and the focus on mitigating one's own deficits rather than accepting them (*Smedema & McKenzie, 2010*). We also believe that focusing on disability-related topics can make it difficult to move out of thinking of oneself as "disabled" and into full inclusion in society. Yet another study showed that access to the Internet provides people with disabilities with the possibility to communicate with friends, spend and enjoy leisure time (*Shpigelman & Gill, 2014*). As explained by the compensation model, if someone is socially inactive off-line, he or she benefits more from using the Internet (*Lathouwers, de Moor & Didden, 2009*).

Downloading music files and software without a license and viewing sites with sexual content were the most common risky behaviours among our students with VI. These results are consistent with the preliminary report which indicated the same tendency (*Wrzesińska, Tabała & Stecz, 2016*). It should be added that one in five students with VI had experiences with online pornography and this prevalence is nearly twice as high among Polish non-VI peers (*Pyżalski et al., 2019*). The third hypothesis regarding gender disparities in the frequency of risky behaviours was confirmed. Generally, male students engaged in all risky behaviours more often than female students, including downloading software illegally and hacking. Moreover, the male participants of this study viewed websites with sexual content more often than their female counterparts. This results are consistent with a pilot study (*Wrzesińska, Tabała & Stecz, 2016*).

The aspect of sexuality seems to be worth analysing in the context of young people with VI. As sexuality starts to play an important role in puberty, it is natural that adolescents search for knowledge, also on the Internet, and develop their own norms and attitudes towards sex (*Naezer, 2018*). At the same time, however, there appears the risk of exposure to specific manifestations of violence, sexting, negative stereotypes, or dehumanization (*Stanley et al., 2018*). From this point of view, young people with VI, who experience difficulties in forming intimate relationships in real life and try to compensate for it in the virtual world, can be more exposed to sexual online violence. Still, it seems understandable that youth with VI search for sex and sexuality-related information, especially given that they have diminished possibilities to acquire knowledge by observation or contact with peers, compared to young people without sensory deficits, and when sex education at home or school is not enough or is not tailored to their specific needs (*Czerwińska, 2019*).

Our study showed that more participants with VI were perpetrators of threatening or offending others on the Internet than students without disabilities from Polish schools (*Pyżalski et al., 2019*). Before the study we hypothesized that respondents with VI are more likely to be victims than perpetrators of electronic aggression. Although the results were not statistically significant, the highest scores were observed for witnessing electronic aggression, and there were more victims than perpetrators of online aggression. There were also no statistical differences between genders in this respect, but boys experienced being a witness, a victim, or a perpetrator of electronic aggression more often than girls in our study group. Although our results are not the basis for generalization, a particular attention should be paid to online violence among young people with a problem of visible bodily injury, disability or a serious health problem. Acts of aggression include different

phenomena such as unpleasant comments in one-to-one online communication, unpleasant public comments, revealing the secrets of particular people in online communication, the dissemination of disgraceful information concerning a person. Moreover, the online material has some additional characteristics namely durability, copyability and the possibility of access by an invisible audience. The published information widely available in this way increases the risk of losing control regardless of the will of those involved (including the original perpetrators). It is recommended that we should educate people with disabilities by raising awareness of the mechanisms of building their own online image and forming the right attitudes of emotional resilience towards other people (*Plichta, Pyżalski & Barlińska, 2018*).

Furthermore, communication with a person promoting violence and the mean time of online activity are significant predictors of the likelihood of being a witness, a victim, or a perpetrator of electronic aggression. Our study also showed that the more time students spend on the Internet during school-free days, the more likely they are to become a witness or a victim. The risk of becoming a perpetrator of electronic aggression depends on the online time during schooldays. It was also revealed that students with VI having occasional online contact with a person promoting violence become three times more often a witness or a victim, and four times more often a perpetrator of electronic aggression. They are even five times more likely to become the victims of electronic aggression if they come into contact with such a person often or very often, compared to those who have never communicated with the perpetrator of aggression. The constant access of perpetrators to their victims on Internet is the mechanism that makes electronic aggression even more destructive than the traditional one (*Pyżalski, 2012*). Many young people, mostly because of using SNS, are "always connected" and they are constantly exposed to harmful content and behaviours towards them. Even if they decide to be offline, negative information about them could be further disseminated over the Internet (*Pyżalski, 2012*).

Our findings are not free from limitations. The first limitation is the cross-sectional design that makes some causal reasoning impossible. Furthermore, a self-administrated survey was used to enable anonymous participation in the study. Respondents were supported by researchers and teachers were not present during the process of filling in questionnaires, which was meant to enhance honesty of the answers provided. Our intention is to repeat the study among young people with VI attending regular schools and take into consideration their parents' and teachers' opinions. Moreover, the results of our research are difficult to compare with the results of other studies of peers without or with disabilities. This is due to the lack of validated diagnostic tools dedicated to people with different disabilities. It is recommended to develop universal study protocols for students with and without disabilities to increase the chances of comparing the results. Finally, qualitative research among different groups of youngsters could be provided in the next step of future research. It will provide a better opportunity to develop diagnostic solutions regarding risky behaviours among participants with and without disabilities. In this way, we believe, our research will contribute to identifying the needs and challenges faced by groups at the risk of social exclusion and, as a result, become a starting point for

the development of academic, educational and social initiatives, as well as media literacy, meant to reduce the extent of digital divide and ensure a better quality of life for people with disabilities.

## CONCLUSION

Cyberspace with its technological affordances has enabled new proximity dynamics in user-to-user and user-to-content interactions, as argued by the MPA (*Kopytowska, 2020*). Allowing users to transgress time and space boundaries it has considerably extended possibilities of participation and agency. The safe use of the Internet enables and facilitates information exchange, supports the learning process, provides an opportunity for self-expression and communication with people having similar interests. Yet, however beneficial in educational terms and in the context of social participation using the Internet is, it is also burdened with the risk of exposure to various types of online aggression and otherwise harmful content.

Considering the above factors when teaching digital competencies and media literacy we should place particular emphasis on constructive and safe media use among young people with VI. Safety-related awareness-raising activities should address problems of cyberbullying, electronic aggression, including "digital dating abuse" (*Hinduja & Patchin, 2020*), as well as other online risky behaviours. Sexual health education is also necessary to protect adolescents with VI against sexual violence and ensure proper development within this sphere of psychosocial functioning (*Hafiar et al., 2020*). Not only students, but also parents, caregivers, teachers and health educators should be confronted with possible risks associated with cyberspace and provided with knowledge about preventive resources, strategies and tools adjusted to the special needs of people with disabilities (*Symons et al., 2017*). Another important and protective factor is developing social skills and enhancing interpersonal contacts in real life to minimize the need to compensate for loneliness on the Internet.

### Funding

This work was supported by the Gambling Problem Solving Fund, Ministry of Health of the Republic of Poland [Grants number: 501/5-127-03/501-81-130]. The funders had no role in study design, data collection and analysis, decision to publish, or preparation of the manuscript.

### Grant Disclosures

The following grant information was disclosed by the authors:
Gambling Problem Solving Fund, Ministry of Health of the Republic of Poland: 501/5-127-03/ 501-81-130.

### Competing Interests

The authors declare that they have no competing interests.

## Author Contributions

- Magdalena Agnieszka Wrzesińska conceived and designed the experiments, performed the experiments, analyzed the data, prepared figures and/or tables, authored or reviewed drafts of the paper, and approved the final draft.
- Kamila Knol-Michałowska conceived and designed the experiments, analyzed the data, authored or reviewed drafts of the paper, and approved the final draft.
- Patryk Stecz conceived and designed the experiments, performed the experiments, analyzed the data, authored or reviewed drafts of the paper, and approved the final draft.
- Monika Kopytowska conceived and designed the experiments, analyzed the data, authored or reviewed drafts of the paper, and approved the final draft.
- Katarzyna Binder-Olibrowska conceived and designed the experiments, performed the experiments, analyzed the data, authored or reviewed drafts of the paper, and approved the final draft.

## Human Ethics

The following information was supplied relating to ethical approvals (*i.e.*, approving body and any reference numbers):

The project was approved by the Bioethics Committee of the Medical University of Lodz (473 RNN/802/14/KB).

## Data Availability

The data are available at Zenodo: Magdalena Wrzesińska, Kamila Knol-Michałowska, Patryk Stecz, Monika Kopytowska & Katarzyna Binder-Olibrowska (2021). Internet risky behaviours among youth with visual impairment. https://doi.org/10.5281/zenodo.5515310.

## Supplemental Information

Supplemental information for this article can be found online at http://dx.doi.org/10.7717/peerj.12376#supplemental-information.

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
