# Peer review of "Internet risky behaviours among youth with visual impairment"

_PeerJ, doi:10.7717/peerj.12376_

## Round 0.1 · original submission · Major Revisions

While I seek to find a balance between the terms Major and Minor, I am left to suggest that you seriously consider addressing comments by both reviewers, especially the reviewer who suggests Major Revisions moving forward, and whether that may be readily translatable at your end into Minor revisions which you will thereby conduct and resubmit for reconsideration. Thank you for letting us review your important paper. I look forward to your revision decision moving forward.

Reviewer 1 ·

Basic reporting

The paper is well written.

Experimental design

The paper investigates the risks faced by visually impaired adolescents when accessing the Internet. The study protocol is robust.

Validity of the findings

The findings are factual and useful.

Additional comments

The study is useful and the paper is well written. You may consider more thorough statistical analysis in your future papers.

·

Basic reporting

Clear English: The article is mostly written in understandable English; however, there are a number of minor glitches that I've identified under Section 4. General comments and provided suggestions for edits.

Literature: The article does give some background to the issues examined in it; however, there are some areas where more background information would be useful. In particular, you could describe how the Internet use of blind and partially-sighted youth differs in practical terms from youth without visual impairments (e.g. screen reading software, large print, zooming functions?) This would be useful for those less familiar with these technological solutions and their limitations. In addition, there are some definitional issues etc. that I've commented on specifically in Section 4 below.

Data, structure, figures: The data have been shared and the tables and general structure of the article seem fine.

Self-contained: Yes, the article is self-contained. Objectives stated instead of hypotheses - from my point of view, this is fine.

Reference list: There are quite a few sources in Polish. While I'm not sure about this journal's style requirements for these situations, it would be helpful for readers unfamiliar with Polish to have their names translated into English in brackets after the original Polish name.

Experimental design

Original research: The article does report original primary research.

Research questions: Objectives are meaningful but could be a bit more specific than they currently are.

Technical and ethical standard: The technical standard of the article seems fine, and informed consent was obtained. My only reservation is whether the consent process sufficiently took into account the special needs of the participants (see Section 4 below).

Methods described with sufficient detail: This seems to be the case.

Validity of the findings

The data are provided, and conclusions are generally supported by the data. However, the lack of non-VI control group reduces the usefulness of the findings, which the authors have acknowledged in the Limitations. If there is any comparable research on non-VI youth in Poland, comparisons should be made with it. The authors might also recommend conducting a replication study featuring participants with and without VI for the sake of data comparability (this would also be a great study for them to carry out themselves).

Additional comments

Title and terminology in the text: since adolescence is defined by WHO ( https://www.britannica.com/science/adolescence ) as the age range of 10-19 years, whereas "young people" are defined as those who are 10-24 years old, considering that your sample has people up to the age of 24 years, it would be better to use the term "youth" or "young people" instead of "adolescents" in your title and throughout your manuscript where references are made to your own participant group. "Boys" and "girls" could be replaced with "male youth" and "female youth" (as uncountable nouns) to correspond with this terminology (alternatively, "male students" and "female students" are fine and you have already used these at times).

l. 68-69: It would be good to provide brief (a line or two) contextual information about the EU Kids Online project (aims, number of countries and participants, etc.) to enable readers unfamiliar with it to see why it's so relevant to your study

l. 84 It would be good to provide a definition for electronic aggression the first time you introduce it; it would also be helpful if you can explain how you see it as same or different than cyberbullying, for example (with appropriate citations to the literature)

l. 87 "and also about of people aged": some percentage missing here?

l. 90 please clarify what you mean by the term "techno-discursive design"

l. 93 "...in a way that cannot be controlled" - this is quite a sweeping statement, how can you be sure?

l. 96 please define the terms flaming and trolling in brackets

l. 101 after reference endnote [12] might be a good place to split the paragraph, which is a bit long

l. 110-115 it would be good to contextualize the studies by stating where they were conducted

l. 131-132 They may be scarce, but if you are aware of some, it would be good to cite a few here to show what's already been done in this field

l. 145 On lines 152-153 you stated the questionnaires were in Braille or large print - what about the participant information sheet and consent form? Were they in an accessible format? Do you have any reflections on best practice in obtaining consent from young people with visual impairment?

l. 154 "The mean time of Internet sessions..." - did you count activities both on computers and mobile devices? Please state in the text.

l. 156 This line talks about uploading MP3s, whereas line 193 talks about downloading them. Did you actually ask about both and do you think there is some relevant risk related to either activity?

l. 209 "p= Chi2=12.596;" - is something missing or out of place here?

l. 244 "In the ordinal regression model..." - In order not to confuse readers, it might be clearer if you state "In the second ordinal regression model..."

l. 248-251 The sentence "The odds for participants..." is confusing, please check - especially at "were 2.60 (p<.05) compared"

l. 255 "In the next ordinal regression model" -> "In the final ordinal regression model"

l. 257-259 "The odds for participants..." again a confusing sentence, please check

l. 273 "competences" -> "competencies"

l. 278-280 "Previous study revealed that using online chat..." - curious finding, do you have any speculation as to why seeking disability-related information or participating in support groups lowered well-being, when they are in principle supposed to do the opposite?

l. 330 "teaching digital competences" -> "when teaching digital competencies"

Limitations: It's good that you have noted the limitation about the lack of comparable data here - nevertheless, I wonder if there are no studies conducted with Polish youth without VI using any of the same variables as the present study. If there were any, comparison with the present study's findings would be most useful for giving some indication as to whether adolescents with VI are at an increased risk, decreased risk, or equal risk of suffering negative consequences from their Internet use when compared with adolescents without VI.

l. 396 "bahaviours" -> "behaviours"

l. 424 "Waszyngton" -> "Washington"

p. 21, 23, 25 "Each data point activities faced by students" - confusing, consider rewording

·

Basic reporting

- Clear and unambiguous, professional English is used, with some exceptions which can easily be corrected.
- The description of the general state of research could be more detailed. In this regard, important terms should be better defined.
- The manuscript would benefit from a more thorough structuring.
- No hypotheses are mentioned.

Experimental design

- The investigation seems to have been conducted rigorously and to a high technical and ethical standard, but some important details regarding the procedure of data collection are missing.
- No experiment is conducted. It could be described as quasi-experimental design, but data is not analysed as such.
- Research question should be stated more clearly and be integrated better into the general state of research part.
- Methods are not described in sufficient detail.

Validity of the findings

- Discussion and conclusion would benefit from a more thorough link with the current state of research.
- For several arguments in the discussion, it is not clear how they relate to the reported data.

Additional comments

The authors address the question how young participants with visual impairments engage in different types of internet risk behaviors. The topic is of great importance and interest for the research community as well as for practitioners. The data set is quite impressive, with a total sample of 490 students answering a self-administered questionnaire. In addition, the manuscript is in most parts well written and understandable.

However, the manuscript also contains various elements that can be criticized and have potential for improvement. The most important improvement would imply a stronger focus on a specific research question which is based on a clear research gap. This would also include the formulation of directed hypotheses at the end of the theory section of the manuscript. I would have expected a clear statement of the research gap and the research question at the beginning of the manuscript, followed by a more detailed and better organized review of the state of the art in the domain. In this regard, the line of argument could be strengthened with a clearer focus on the research question and the specific user group. Additionally, terms and concepts are often mentioned very briefly and are not defined thoroughly (e.g. flaming, trolling, incivility), which is an issue for the understanding of the study details and the integration of the findings into the general state of knowledge. The review of previous research should be more detailed in order to be able to understand the knowledge this study adds to the general state of research.

The method section could be structured in sub-headings (e.g. participants, measures and instruments, procedure, data analysis). It was not clear to me why the authors refer to the term ‘study group’. This somehow implies that there is also some sort of control group – which seems not to be the case (unfortunately) in this study (as compared to the ‘pre-study’ in which two user groups were compared [30]). The measures used in this study should be described in more detail. All the questions asked should be mentioned, including the answering options - other researchers should be able to replicate this study based on the information presented in the method section.

As no hypotheses have been formulated, it is quite difficult to comment on the data analysis section.
The discussion section reads in large parts as a repetition of the results section and the integration of the findings into the larger context of the general state of research could be improved. Additionally, I would suggest not to end the manuscript with the limitations (which should be reported in the discussion section).

---

## Round 0.2 · Minor Revisions

A reviewer has been kind enough to reveal where the minor issues are so that authors may be expeditious in their undertaking here. Please review and make any noted, necessary changes for resubmission. Thank you.

·

Basic reporting

The authors have successfully resolved several issues identified by each reviewer. However, the manuscript appears not to have been thoroughly proofread, copyedited, or edited for correct grammar. I've identified all of these issues in Section 4 below. If the authors resolve those issues I think it should be publishable then.

Experimental design

There are minor details that need to be added - see Section 4.

Validity of the findings

The article is in much better shape after the previous edits with regard to this area.

Additional comments

Here are my detailed line-by-line suggestions and questions:

l. 50 I see "risky Internet use" was changed to "risk Internet use". The original wording is more grammatically correct, please revert back to that.

l. 71 Missing a space between "youth" and "with"

l. 77-78 "provide a high quality, independent and comprehensive research findings" -> "provide high quality, independent and comprehensive research findings"

l. 80 "an integrated" -> "an integrated" (delete 1 excess space)

l. 85-87 "Social networking services (SNS), video/audio conferencing tools, games, movie watching and listening to music are used" -> "Using social networking services (SNS) and video/audio conferencing tools, playing games, watching movies and listening to music are done"

l. 118-120 For clarity, please cite the source immediately after the sentence giving the definition of electronic aggression. Just wondering - the definition looks exactly like the textbook definition of cyberbullying. If you consider electronic aggression and cyberbullying to be the same thing, it might be good to state this to link your work to the broader cyberbullying research.

l. 121 "study showed" -> "study showed"; "off-line about" -> "off-line, about" (remove 1 excess space in both cases, add comma in the latter)

l. 122 "person aged" -> "person aged" (remove excess space)

l. 123 "of an electronic aggression" -> "of electronic aggression" (electronic aggression is an uncountable noun, hence no article)

l. 123 "about 14% of older students have received sexual messages and 7% of participants have been perpetrators of cyberbullying or hate speech." -> Clarify which study's participants were described

l. 129 "media content" -> "media content." (add missing full stop)

l. 129-131 The added sentence added isn't quite clear yet. It still needs a brief definition of "discursive power" (maybe in brackets after the term) and "automatically equate visibility or popularity" seems to be missing the thing that they're being equated with?

l. 133 The word "sometimes" should be before the word "encourage" ... or should it be ", sometimes in an uncontrollable way" ?

l. 140 "harassment)[15]" -> "harassment) [15]" (Add missing space)

l. 175-176 Add missing line break between paragraphs

l. 177 "Youth with physical disability is excluded" -> "Youth with a physical disability are excluded"

l. 179 "for Internet" -> "for Internet" (delete excess space)

l. 189 "to see better the content" -> "to see the content better"

l. 223 "youth with" -> "youth with"; "Another preliminary" -> "Another preliminary" (delete excess spaces)

l. 231 "risk behaviours" -> "risk behaviours" (delete excess space)

l. 232 "poor-sighed. [32, 34]. Rresearch" -> "poor-sighted [32, 34]. Research" (delete excess spaces and full stop, correct spellings)

l. 234 "such as: gender," -> "such as gender," (no comma after "such as")

l. 241 "The following hypothesis " -> "The following hypotheses " (use plural form because more than one hypothesis is stated)

l. 248 "perpetrators of" -> "perpetrators of"

l. 249-250 "all types of electronic aggression (victim, perpetrator, witness)" -> "all forms of engagement in electronic aggression (victim, perpetrator, witness) "

l. 309 "to International" -> "to the International"

l. 319-320 Add missing line break between paragraphs or combine into the same paragraph, as appropriate.

l. 324 "survey. The" -> "survey. The" (delete excess space)

l. 326 "why 490" -> "why 490" (delete excess space)

l. 327 "recruited" -> "recruited." (add missing full stop)

l. 329-330 "known as the sensitive data" -> "considered sensitive data"

l. 330-331 Add missing line break between paragraphs or combine into the same paragraph, as appropriate.

l. 351-443 Add missing line breaks between paragraphs or combine into the same paragraph, as appropriate.

l. 354-355 "the sociodemographic variables such as: gender, age (birth year) and type of school were collected." -> "questions on sociodemographic variables included gender, age (birth year) and type of school."

l. 357-358 "The time of the online activities on computer and mobile devices was estimated based on participants’ declarations." -> "Time spent on online activities on computers and mobile devices was estimated on the basis of participants’ declarations."

l. 366 "(downloading the software" -> "(downloading software"

l. 373 "performing three roles: perpetrators, victims, and witnesses." -> "with three roles: perpetrators, victims, and witnesses." (being a victim or witness aren't really "performed")

l. 377 "comprised five-point Likert scale" -> "used a five-point Likert scale"

l. 438 "behaviors" -> "behaviours" (so far the article used UK spelling - check consistency throughout the article).

l. 449 "and almost" -> "and almost" (delete excess space)

l. 451 "students both" -> "students both" (delete excess space)

l. 457-459 Should the Chi-Square figures be reported for differences between male and female students?

l. 460 "to almost" -> "to almost" (delete excess space)

l. 491 "downloaded the software" -> "downloaded software"

l. 495-496 "and younger students more often resorted to hacking than older students [(N=8, 5.3% vs N=10, 3.1%; Chi2=5.652; p=0.059)]." - since the Chi-Square was only marginally significant and the difference was only two students per group, this might be better to leave out.

l. 499 "N=1.3%," - something's missing here

l. 500-502 Again, very small differences in absolute figures. Consider including Chi-Square figures if significant and deleting if non-significant.

l. 507 "threating" -> "threatening others"; also, what behaviors does "offending" include? Or should it be "threatening or offending others"?

l. 516 "downloading the software" -> "downloading software"

l. 518 "on schooldays" -> "on schooldays"

l. 560 "often compared to" -> "often than"

l. 593 "in the adulthood among" -> "in adulthood among"

l. 594 "group. The" -> "group. The"; "for social" -> "for social" (delete excess spaces)

l. 595 "which confirms our assumption" - with regard to which hypothesis?

l. 610 "high comparing to Polish non-VI peers " -> "as among Polish non-VI peers"; "The hypothesis regarding gender disparities" - you numbered the hypotheses in the introduction, so it'll be good to state the number of the hypothesis in question here.

l. 612 "downloading the software" -> "downloading software"

l. 613-614 "the males examined viewed websites with sexual content more often than females." -> "the male participants of this study viewed websites with sexual content more often than their female counterparts."

l. 676 "making intimate relationships" -> "forming intimate relationships"

l. 679 "especially that" -> "especially given that"

l. 682 "threating or offending" -> "threatening or offending others"

l. 686 "statistically relevant," -> "statistically significant"

l. 687 "are also" -> "were also"; when you say that there were no statistical differences between genders, and yet you describe such a difference, does this mean that there was a difference in the sample but it was not generalizable to the population due to non-significant chi-square tests? Please clarify.

l. 691 "eg. unpleasant" -> "such as unpleasant"

l. 694 "i.e." -> ", namely"

l. 695 "by invisible audience" -> "by an invisible audience"

l. 726-727 "self-administrated form of survey was used in order to enable anonymity." -> "a self-administrated survey was used to enable anonymous participation in the study."

l. 733-734 "the universal study protocols" -> "universal study protocols"

l. 752 "digital competences" -> "digital competencies"

References: Please check once more to ensure correct copyediting in keeping with the journal's style

Overall: Please copyedit and proofread the manuscript carefully before submitting the revised version. It may be helpful to view the file without tracking turned on before submitting it, or even print out the manuscript for final inspection, to see minor copyediting issues more clearly.

---

## Round 0.3 · accepted · Accept

Congratulations and thank you for addressing most issues pointed out by reviewers. Note that there are still typos in the document, such as "threating" where it should be "threatening" and "bulling" instead of "bullying". I have noted these along with recommendations to PeerJ staff but trust that they are easily and readily resolved for ready publication.